# Biogenic carbonate mercury and marine temperature records reveal global influence of Late Cretaceous Deccan Traps

Kyle W. Meyer [1]*, Sierra V. Petersen[1]*, Kyger C Lohmann[1], Joel D. Blum[1], Spencer J. Washburn[1], Marcus W. Johnson[1], James D. Gleason[1], Aaron Y. Kurz[1] & Ian Z. Winkelstern[1,2]

The climate and environmental significance of the Deccan Traps large igneous province of west-central India has been the subject of debate in paleontological communities. Nearly one million years of semi-continuous Deccan eruptive activity spanned the Cretaceous-Paleogene boundary, which is renowned for the extinction of most dinosaur groups. Whereas the Chicxulub impactor is acknowledged as the principal cause of these extinctions, the Deccan Traps eruptions are believed to have contributed to extinction patterns and/or enhanced ecological pressures on biota during this interval of geologic time. We present the first coupled records of biogenic carbonate clumped isotope paleothermometry and mercury concentrations as measured from a broad geographic distribution of marine mollusk fossils. These fossils preserve evidence of simultaneous increases in coastal marine temperatures and mercury concentrations at a global scale, which appear attributable to volcanic $CO_2$ and mercury emissions. These early findings warrant further investigation with additional records of combined Late Cretaceous temperatures and mercury concentrations of biogenic carbonate.

[1] Department of Earth and Environmental Sciences, University of Michigan, 1100N. University Ave., Ann Arbor, MI 48109, USA. [2] Geology Department, Grand Valley State University, 1 Campus Drive, Allendale, MI 49401, USA. *email: kymeyer@pdx.edu; sierravp@umich.edu

Large igneous provinces (LIPs) have recently garnered renewed attention at critical extinction intervals and climate perturbations throughout the geological record due to the application of mercury concentrations, [Hg], and isotopic compositions[1–6]. Volcanically emitted gaseous elemental mercury (Hg⁰) represents the largest natural source of Hg to the atmosphere[7,8]. A short atmospheric residence time (~1 year), combined with an assumed modern-equivalent interhemispheric mixing time of ~1 year, allows for global distribution of volcanically emitted Hg⁰ and produces a traceable environmental fingerprint of LIP activity. The relative role of the Deccan Traps LIP of Western India alongside the Chicxulub impact event at the Cretaceous–Paleogene (K–Pg) boundary, in terms of potential contributions to the concurrent mass extinction, has been subject to longstanding debate[9–12]. Records of environmental Hg reconstructed from deep sea sediment cores have provided insight into the timing and scale of this event[1,2], but lack a clear linkage to a climate and/or a biotic response and may be susceptible to misinterpretation in sedimentary records where Hg may be dominantly sulfide-hosted (e.g., in black shales)[13].

Refined geochronology of the Deccan Traps LIP has indicated that >$10^6$ km³ of flood basalt lava was extruded from semi-continuous eruptions over ~1 Myr, with roughly 25–50% of that material being emplaced in the 250 kyr prior to the 66.0 Ma K–Pg boundary[14–17]. Numerous studies have suggested that more frequent and sustained eruptive activity occurred prior to the K–Pg boundary, with large hiatuses between eruptions afterward based on observations of paleosol red bole horizons[14–18]. Therefore, it can be assumed that the Deccan eruptions prior to the Chicxulub impact event could have provided more sustained input of $CO_2$ and other volatiles to the atmosphere compared to later eruptions. Estimates of total $CO_2$ released from the Deccan Traps range from 2800 to 21,600 Pg C, and could have produced a global warming climate response (<1–6 °C) for over 400 kyr following the initiation of the eruptions[19].

We hypothesized that marine mollusk fossil carbonate would simultaneously record both coastal marine temperature responses and varying [Hg] signals associated with the release of $CO_2$ and Hg⁰ from the Deccan Traps, respectively. The conceptual model for how enhanced atmospheric Hg flux is recorded in the biogenic carbonate of fossil bivalves (Fig. 1) involves the photochemical oxidation of Hg⁰ in the atmosphere to Hg(II), subsequent dissolution in rainfall or sorption to aerosols followed by wet and/or dry deposition to the oceans[20,21]. Upon entering the marine environment, Hg(II) becomes highly reactive with particulate organic matter and is the dominant species of Hg to undergo uptake by phytoplankton[20–22]. Benthic and sessile bivalves are consumers of phytoplankton and are known to accumulate significant Hg (II) in their soft tissues[23]. The presence and accumulation of Hg in the shells of modern freshwater bivalves is less well studied but has been documented as a potential bioindicator of environmental water quality[24]. Other studies have produced archival records of [Hg] in biominerals (such as hydroxyapatite in bone and biogenic carbonate of avian eggshells) from the Holocene[25,26], but we have reconstructed the first deep time record of [Hg] preserved in fossilized biomineral remains. We measured [Hg] in biogenic carbonate ($CaCO_3$) paired with carbonate "clumped" isotope compositions ($\Delta_{47}$)[27,28], an environmental indicator of past marine temperatures[29]. We observe an apparent global signal of abruptly increased temperatures and elevated [Hg] in the same specimens, prior to the impact event at the K–Pg boundary, but aligning with the onset of Deccan volcanism. This records a quantifiable climate and environmental influence from LIP volcanism near this key extinction event.

## Results and discussion

**Results summary**. We present marine temperatures from clumped isotope ($\Delta_{47}$) compositions and [Hg] measured in fossil bivalves from Seymour Island, Antarctica and Moscow Landing, Alabama (Fig. 2). We also report the variability in [Hg] of mollusk biogenic carbonate from a collection of globally distributed localities (Fig. 3). The specimens collected from these sites represent a range of time intervals across the Late Cretaceous, the Pleistocene (to serve as a preindustrial background) and modern (to provide a measure of [Hg] in areas of modern background and point-source contamination from legacy industrial Hg pollution; Fig. 4). Details regarding these sites and the specimens collected can be found in Supplementary Note 1.

**Time series of mercury concentrations and marine temperatures**. In both the Antarctica and Alabama K–Pg boundary records (Fig. 2), we observe a distinct correspondence between [Hg] and $\Delta_{47}$-derived coastal marine temperatures as analyzed in fractions of the same aliquot from any given fossil specimen. Temperature estimates vary at both sites by as much as 15 °C over the entire study interval, with an abrupt warming (approximately 5–12 °C) within 250 kyr prior to the K–Pg boundary[27,28,30]. This abrupt warming interval is contemporaneous with the onset of Deccan volcanism[14–17], which is highly suggestive that this climate forcing was driven by the emission of volcanic $CO_2$, as implied in previous studies (and references therein)[19,27]. In the Antarctic section, we observe peak [Hg] of 17.1 ($n = 3$, $2\sigma = 0.6$ ng g⁻¹), whereas the Alabama section has higher peak [Hg] levels of 42.0 ng g⁻¹ ($n = 3$, $2\sigma = 3.0$ ng g⁻¹) (Fig. 2). We note that the section in Alabama lacks the same degree of chronological constraint as the Antarctic section[27,28], and is therefore plotted in terms of stratigraphic position instead of absolute ages. The Antarctic section also records elevated [Hg] ~1.5 million years earlier, prior to the onset of the bulk of Deccan volcanism, which may correspond to a known earlier, smaller phase of eruptive activity in Western India described by transitional flow units up to 100 m thick, tentatively dated to $67.5 \pm 0.6$ Ma[31]. Elevated [Hg] levels increasing with time approaching the K–Pg boundary were also observed in a small number of specimens, constrained to the Late Maastrichtian by strontium isotope ($^{87}Sr/^{86}Sr$) stratigraphy and limited biostratigraphic studies of the region (please refer to Supplementary Data File 1).

**Mercury concentrations of modern mollusk biogenic carbonate**. Since mollusk shells have not been used previously in reconstruction of geologic Hg emissions levels, we analyzed modern shells from various locations to investigate the extent to which bivalve [Hg] correlates with enhanced environmental [Hg]. Based on our results presented in Fig. 4, we propose a "modern anthropogenic background" of [Hg] in biogenic carbonate at a threshold of about 5 ng g⁻¹ defined from the range of measured [Hg] values of specimens from all modern uncontaminated sample localities (0.2–4.3 ng g⁻¹). Each modern uncontaminated locality (Providence, Rhode Island; Lake Tahoe, California; Spectacle Island, Massachusetts) is subject to substantial anthropogenic influence (e.g., waterfront industrial plants, active shipping lanes, recreational boating, and urban effluent). Gasoline, diesel, and liquefied petroleum gas have been shown to contain [Hg] from 180 to 1200 ng L⁻¹, with automotive engine exhaust [Hg] ranging from 1.5 to 26.9 ng m⁻³, depending on the fuel source[32], and serve as potential inputs of Hg to the habitats of specimens collected. In addition, all modern sites are subject to a three to fivefold increase in Hg deposition from the atmosphere compared to before the Industrial Revolution[33–35].

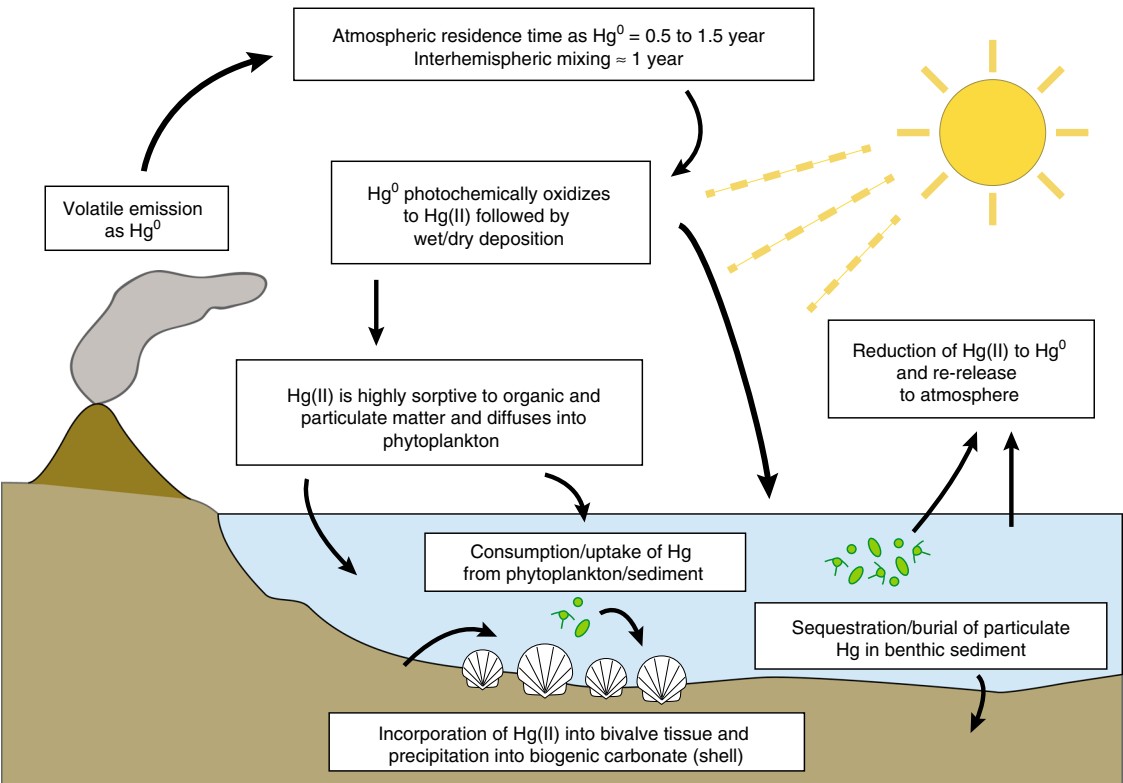

**Fig. 1** Conceptual model of volcanogenic Hg emission and subsequent incorporation into marine biota. Gaseous elemental mercury ($Hg^0$) is released in addition to other volatiles (e.g., $CO_2$), where in the atmosphere it is photochemically oxidized to Hg(II) and either adsorbed to particulate and/or organic matter, reduced and rereleased back to the atmosphere, or incorporated into phytoplankton through various uptake pathways. Benthic, sessile, filter-feeding marine mollusks can then bioaccumulate Hg(II) from either the consumption of algae or from the sediment directly.

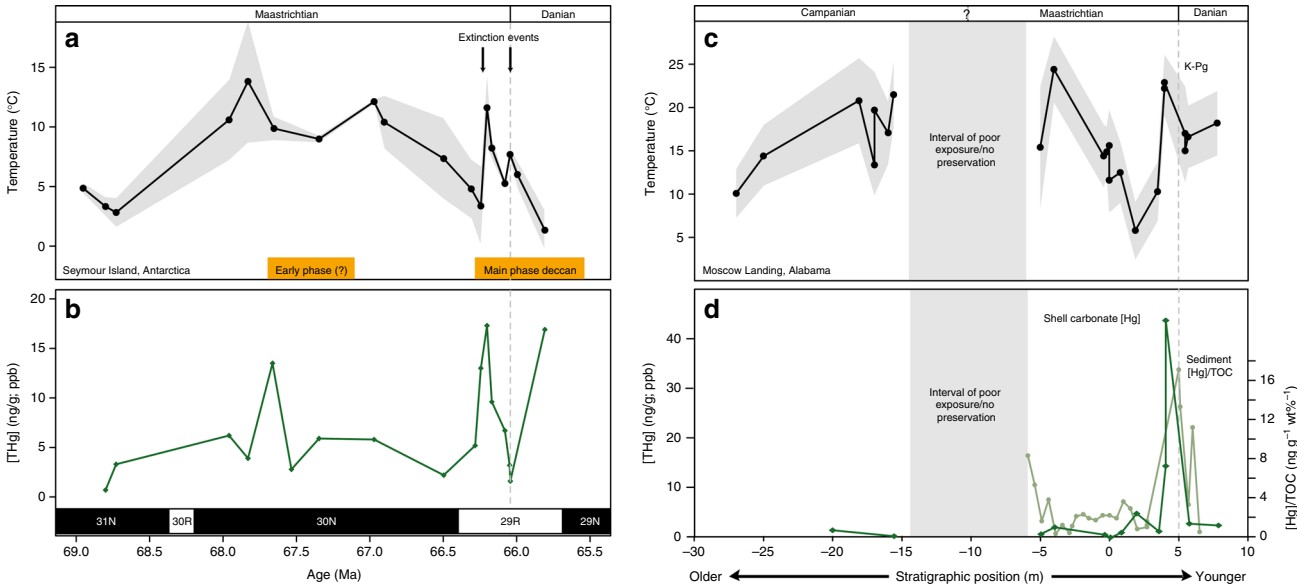

**Fig. 2** Coastal marine temperature and mercury concentration time series. $\Delta_{47}$-derived coastal marine temperatures (**a**, **b**) and [Hg] records (**c**, **d**) for Seymour Island, Antarctica[27] and Moscow Landing, Alabama[28]. The age model for Seymour Island was derived from combined biostratigraphy, magnetostratigraphy[55], strontium isotope chemostratigraphy[56], and the presence of the Ir anomaly to demarcate the K–Pg Boundary[57], as described in previous studies[27]. The K–Pg boundary at Moscow Landing is an unconformable contact, and the entire section is likely only a portion of what is preserved at Seymour Island. Nonetheless, at both sites we observe a close coupling between elevated [Hg] and temperatures immediately prior to the boundary. Uncertainties on any single sample [Hg] value are, on average, ±11% of the measured [Hg] value and often too small to depict (please refer to Methods). Uncertainties on the $\Delta_{47}$-derived marine temperatures are calculated at 1 S.E. and depicted as shaded regions adjacent to enveloping the data. The extinction events depicted at Seymour Island, Antarctica are based on macrofossil occurrence patterns from that locality[55,58]. These temperatures all compare well to existing $\delta^{18}O$-derived temperature proxies[55,59,60]. Source data are provided in Supplementary Data Files 1, 2 and 3.

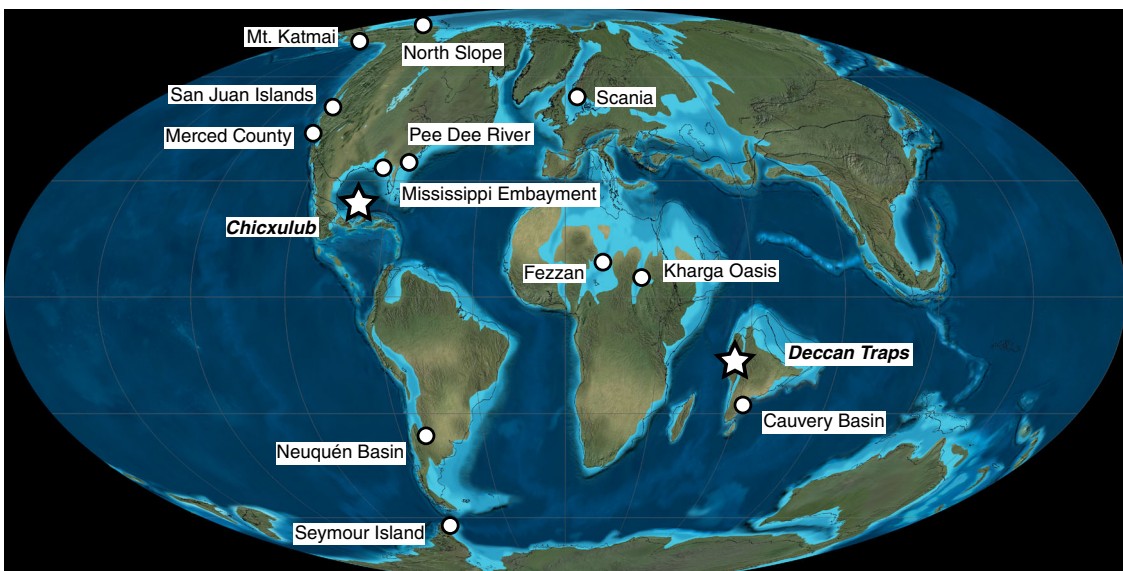

**Fig. 3** Late Cretaceous paleogeography and relevant sample regions of the study. Depicted is a paleogeographic reconstruction at the time of the K–Pg Boundary, labeled with specimen sample regions. Individual regions often constitute several localities where sampling occurred either historically or by the authors. Additional specimen details can be found in Supplementary Note 1. Locations of the Deccan Traps large igneous province and the Chicxulub impact event are demarcated with star-shaped symbols. The global paleogeography Mollweide projection base map of the K–Pg Boundary is used with permission from the license holder © 2016 Colorado Plateau Geosystems, Inc[61].

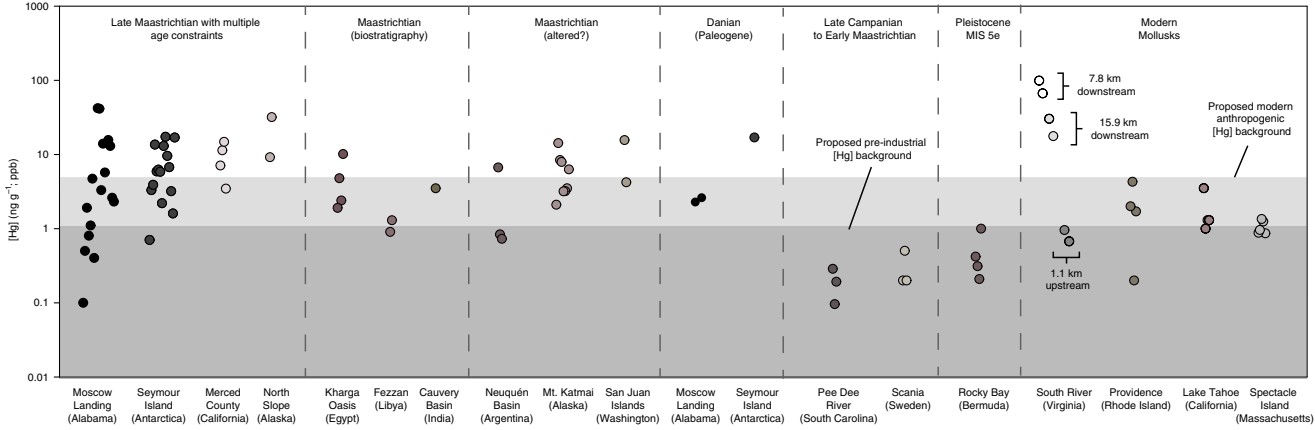

**Fig. 4** Modern and ancient mercury concentrations by sample region. Measured [Hg] values in ancient and modern mollusk shells by sample region divided temporally and by quality of age constraints, plotted on a log scale. Samples from Moscow Landing, Seymour Island, Merced County, and North Slope regions all are Late Maastrichtian in age and are constrained by some combination of biostratigraphy, magnetostratigraphy, strontium isotope stratigraphy, and/or the presence of the "iridium layer." Specimens from Kharga Oasis, Fezzan, and Cauvery Basin are acknowledged to be from the Middle to Late Maastrichtian, but only have biostratigraphic age constraints. Mt. Katmai, San Juan Islands, and Neuquén Basin sample regions are acknowledged to be Maastrichtian in age, but based on $\Delta_{47}$ and/or $^{87}Sr/^{86}Sr$ values are possibly diagenetically altered. Pee Dee River, Scania, and Rocky Bay sample regions are known to exist well outside the timing of the Deccan Traps eruptions, exhibit negligible [Hg], and the range of values of these specimens are used to define a "Pre-Industrial background" with [Hg] ≤ 1.3 ng g$^{-1}$. For comparison, modern uncontaminated mollusk samples from Providence, Lake Tahoe, and Spectacle Island define a "Modern anthropogenic background" with [Hg] ≤ 5.0 ng g$^{-1}$. Lastly, the legacy Hg-contaminated South River site depicts samples from various lengths along the stream corridor relative to the point source of Hg contamination, and permits a first-order comparison to the elevated Maastrichtian fossil [Hg] values associated with the Deccan Traps. Uncertainties on any single sample [Hg] value are, on average, ±11% of the measured [Hg] value and are too small to depict on this figure (please refer to Methods). Full sample locality details can be found in the Supplementary Note 1. Source data are provided in Supplementary Data Files 2 and 3.

Freshwater bivalve shells were also analyzed from a highly Hg-contaminated region along the South River, Virginia (Fig. 4). From samples collected 1.1 km upstream of the industrial point source, [Hg] values were low, ranging from $0.67 \pm 0.06$ ($2\sigma$, $n = 3$) to $0.95 \pm 0.04$ ng g$^{-1}$ ($2\sigma$, $n = 3$). In comparison, [Hg] levels were much higher downstream and decreased with distance from the point source, ranging from $66.5 \pm 1.5$ ($2\sigma$, $n = 2$) to $98.2 \pm 2.9$ ($2\sigma$, $n = 2$) ng g$^{-1}$ at 7.8 km downstream, and from

$17.6 \pm 0.7$ ($2\sigma$, $n = 2$) to $30.3 \pm 1.0$ ($2\sigma$, $n = 2$) ng g$^{-1}$ at 15.9 km downstream. Surface water, suspended material (only for site 1.1 km upstream), and streambed sediments collected concurrently with bivalve shells all show similar patterns to bivalve [Hg] values (see Supplementary Fig. 1), with low values 1.1 km upstream (surface water [Hg] = 0.35 ng L$^{-1}$; suspended material [Hg] = 0.42 ng L$^{-1}$; streambed sediment [Hg] = 0.008 μg g$^{-1}$) and higher values 7.8 km downstream (streambed sediment

[Hg] = 1.54 µg g$^{-1}$) and 15.9 km downstream (streambed sediment [Hg] = 3.68 µg g$^{-1}$) from the point source[36,37]. Surface water and suspended material concentrations measured at proximal downstream sites (5.6 and 13.9 km downstream) were similarly elevated (surface water [Hg] = 7.04 ng L$^{-1}$, suspended material [Hg] = 75.09 ng L$^{-1}$; and surface water [Hg] = 9.69 ng L$^{-1}$, suspended material [Hg] = 92.99 ng L$^{-1}$, respectively)[36]. Furthermore, the bivalve shell [Hg] values from the South River (all a single taxon, *Corbicula fluminea*) are similar to the range of measured [Hg] values determined for the soft tissues collected from live *C. fluminea* along overlapping portions of the stream corridor (~25–120 ng g$^{-1}$ wet weight at 5.6 km downstream and ~21–92 ng g$^{-1}$ wet weight at 13.6 km downstream)[38]. This would suggest a partitioning of Hg(II) between soft tissue and shell carbonate approaching $K_D \approx 1$ in modern freshwater bivalves. Therefore, we have demonstrated that modern bivalves shell carbonate can closely reflect the range of [Hg] from the soft tissues of modern living bivalves, but we also infer that a large increase in sediment Hg concentrations would be necessary to account for the magnitude of elevated [Hg] that we observe for fossil specimens coeval with the Deccan eruptive phases. The similarity between [Hg] values from Late Maastrichtian and modern shell carbonate from the contaminated South River suggests that global Hg loading during the Deccan Traps eruptive interval was of similar magnitude to a Hg contamination-impacted site where fish consumption warnings for humans are in effect[39].

**Further comparisons of modern and ancient mercury concentrations.** We contrast the modern and Late Maastrichtian [Hg] values to those measured from samples constrained to several million years prior to the Deccan Traps eruptive window (Fig. 4, labeled "Late Campanian to Early Maastrichtian") at localities in South Carolina (Pee Dee River) and Sweden (Scania), which showed expectedly low [Hg] (0.2 to 1.3 ng g$^{-1}$). The samples of Campanian age at Moscow landing also demonstrate how similarly elevated temperatures (Fig. 4) do not result in elevated [Hg] concentrations and would contradict an interpretation of increased Hg partitioning into biogenic carbonate simply under elevated environmental temperatures (please refer to Supplementary Fig. 2). We observed similarly low values ([Hg] = 0.2 to 1.0 ng g$^{-1}$) for specimens retrieved from Rocky Bay, Bermuda 115–130 thousand years before present[40], at the height of the previous interglacial interval (Fig. 2, labeled "Pleistocene MIS 5e"). These values suggest that a [Hg] value of <1.3 ng g$^{-1}$ could serve as a "Pre-Industrial/Non-LIP" background for biogenic carbonate samples.

A few sample localities lack the quantitative age constraints to place them definitively within the Deccan Traps eruptive window (Fig. 4), making further interpretations uncertain. We have categorized these (Egypt, Libya, India) as constrained by "biostratigraphy," and they exhibit [Hg] values ranging from 0.9 to 10.1 ng g$^{-1}$. In addition, we found significant [Hg] variability (3.3–15.6 ng g$^{-1}$) in samples from Alaska (Mt. Katmai) and Washington (San Juan Islands) known to be of Maastrichtian age, but these samples had $\Delta_{47}$ and $^{87}Sr/^{86}Sr$ values that would suggest some degree of diagenetic alteration. We do not know whether the level of alteration that affected $\Delta_{47}$ and $^{87}Sr/^{86}Sr$ also modified [Hg] values from these samples as no studies of diagenetic impacts on [Hg] in biogenic carbonate exist (refer to the Supplementary Note 2 and the Methods, for further discussion). Hence, we interpret the results from these sites with caution. However, due to the sensitivity of $\Delta_{47}$ to alteration, it is of significantly lower likelihood that a sample would be well preserved for $\Delta_{47}$, but altered for Hg. We also see no clear

evidence of significant species-specific [Hg] incorporation or partitioning across taxa, and observe predictably consistent [Hg] data from specimens representing millions of years prior to the Deccan Traps eruptions (Pee Dee River, South Carolina, USA) and throughout the Deccan Traps eruptive interval (Moscow Landing, Alabama, USA), which are predominantly of the taxon *Exogyra costata*.

Lastly, we measured the [Hg] of marine sediments taken from the same stratigraphic section at Moscow Landing where fossil shells were collected. When comparing shell carbonate [Hg] to sedimentary [Hg] records at the same site (Supplementary Fig. 3), the two records initially appear decoupled. However, following standard practice for sedimentary Hg studies[1,5], after normalizing sediment [Hg] to the sediment wt% total organic carbon, we observed a distinct pattern of increased sedimentary [Hg] associated with the interval of peak bivalve [Hg] (Fig. 1 and Supplementary Fig. 3), suggesting that bivalve [Hg] values reflect enhanced total sediment [Hg] at this particular site.

When compared to global compilations of faunal changes during the Late Campanian through Late Maastrichtian, we see that our data agree well with observed initial temperature increases (determined using conventional $\delta^{18}O$-derived temperature proxies) from 69.5 to 68 Ma, which also correspond to events of significant evolutionary diversification[41]. These compilations also find clustered extinctions of foraminifera and decreases in species richness in the same interval as our observed abrupt $\Delta_{47}$-derived temperature excursion immediately prior to the K–Pg and during the Deccan Traps eruption interval[41].

The trends in [Hg] data we observe in Late Maastrichtian sample localities indicate a global signal of [Hg] anomalies that align with the onset of Deccan volcanism and are coincident with elevated coastal marine temperatures from carbonate clumped isotope paleothermometry. Therefore, for the first time, we can provide insight into the distinct climatic and environmental impacts on biota living during the Late Maastrichtian Deccan volcanism and across the K–Pg extinction boundary by analyzing a single material, eliminating issues of calibrating age models or stratigraphic disruption due to bioturbation. Our combined investigation of [Hg] in modern and fossil mollusk shells reveals enhanced Hg loading in the Late Maastrichtian at what appears to be a global scale, which is of similar magnitude to that measured at a site of anthropogenic legacy contamination. However, this study represents merely a first attempt at evaluating records of environmental Hg loading in the geological record via biogenic carbonate and would benefit from further analyses of marine biota from other K–Pg boundary intervals across the globe. The chemostratigraphy of [Hg] in fossilized biogenic carbonate may provide unique insights into mass extinctions and climatic perturbations throughout the geological record.

## Methods
**Selected taxa and sample localities.** The samples analyzed for [Hg] in this study include the same specimens analyzed for clumped isotope compositions in previous studies[27,28]. In addition to the aforementioned Antarctica and Alabama samples, we also analyzed fossils for [Hg], $\Delta_{47}$, $\delta^{18}O$, $\delta^{13}C$, and $^{87}Sr/^{86}Sr$ values from Late Cretaceous deposits at localities in Argentina, India, Egypt, Libya, Sweden, and the U.S. states of Alaska, California, and Washington (see Supplementary Note 1, Supplementary Data Files 1–3, and Supplementary Fig. 4). The Argentina[42], Egypt[43,44,45], Libya, and India[44] sample regions range from middle to latest Maastrichtian and were selected as possible records for direct comparison to Seymour Island and Moscow Landing, Alabama[27]. The samples from Washington[46] and Sweden[47,48] are known to be of earliest Maastrichtian age from biostratigraphic constraints[46–49] and were intended to serve as controls with respect to measured [Hg], because the host deposits predate eruptive windows of the Deccan Traps, and thus the samples were expected to have [Hg] values near background. The latitudinal range of sample localities spans from 70ºN (Ocean Point, Alaska; 83–85ºN paleolatitude) to 64ºS (Seymour Island, Antarctica; 67ºS paleolatitude), please see Supplementary Table 1 for detailed information.

Additional samples used in this study, included twenty-eight Cretaceous specimens from eight distinct regions, detailed as follows: six *Glycymerita aleuta* (Mt. Katmai region, Alaska), three *Glycymeris* sp. and two *Gryphaea* sp. (Merced County, California), one *Cyrtodaria* sp. (Ocean Point, Colville River, North Slope, Alaska), two *Arca vancouverensis* (San Juan Islands, Washington), four *Exogyra overwegyi* (Kharga Oasis, Egypt), two *Agerostrea ungulata* (Fezzan Region, Libya and Cauvery Basin, India), three *Pycnodonte vesicularis* (Neuquén basin, Argentina), two *Belemnitella* sp. (Scania, Sweden) and three unidentified bivalves (one from Scania, Sweden and two specimens from the Mt. Katmai and North Slope regions of Alaska, respectively). Specimens were either collected in the field by the authors and/or collaborators, or were loaned courtesy of the University of Michigan Museum of Paleontology (Ann Arbor, Michigan) and the University of California Museum of Paleontology (Berkeley, California). We also have collected modern bivalve specimens from: Sabin Point Park along the estuary of the Providence River in Providence, Rhode Island, with 4 specimens that included *Crepidula fornicata* (MOD-PRO-FORa), *Crassostrea virginica* (MOD-PRO-VIRa), *Geukensia demissa* (MOD-PRO-DEMa), and an unidentified bivalve (MOD-PRO-BIVa); Spectacle Island, Boston, Massachusetts, with four specimens of *Crassostrea virginica* (MOD-SPE-VIRa, MOD-SPE-VIRb, MOD-SPE-VIRc, MOD-SPE-VIRd); Lake Tahoe, California, with four specimens of *Corbicula fluminea* (MOD-TAH-CORa, MOD-TAH-CORc, MOD-TAH-CORd); and the South River, Virgina, with six specimens of *Corbicula fluminea* (MOD-SOU-CORa-1.1, MOD-SOU-CORb-1.1, MOD-SOU-CORa-6, MOD-SOU-CORb-6, MOD-SOU-CORa-14, MOD-SOU-CORb-14). Lastly, we analyzed 4 Pleistocene specimens collected from Rocky Bay, Bermuda, three of which were unidentified bivalves (PLE-RB-UNKa, PLE-RB-UNKb, PLE-RB-UNKc), and one specimen of *Cittarium pica* (PLE-RB-JAPa) collected in a previous study[40].

**Shell sampling and preservation assessment.** All samples were subject to a visual assessment by optical microscopy to observe carbonate fabrics, eliminating samples with obvious recrystallization, and/or sampling away from recrystallized vugs or portions of any given shell specimen. We applied the same criteria for assessing alteration for $\Delta_{47}$ from a previous study[27]. Roughly half the specimens are of the oyster genus *Exogyra*, and were sampled near the ventral margin of the shell over a large enough area to represent at least three distinct layers of ordered carbonate in the shell matrix, presumed to be annual growth bands. Carbonate fragments from the ventral margin were ground by hand using a mortar and pestle. Some smaller individual samples were crushed and used in their entirety (e.g., *Anomia* and *Cyrtodaria*).

**Mercury concentration preparation and determination.** With no established analytical standard for the determination of [Hg] in carbonate samples, we selected a variety of carbonate reference materials to use as in-house standards and to propose as community-wide standards. We compared NIST SRM-88b (dolomitic limestone quarried near Skokie, Illinois, USA), NBS-20 (Solnhofen limestone, Germany; exhausted in terms of commercial availability), USGS COQ-1 (carbonatite from the Oka complex, Lake of Two Mountains, Canada), IAEA-B-7 (limestone collected near Maiella, Abruzzo, Italy), Carrara marble (Italy), and in-house standard LV-3 (limestone from Lake Valley Formation, New Mexico) to two reference materials of known [Hg] values, NRC MESS-3 (Beaufort Sea marine sediment; 91 ± 9 ng/g, certified value) and USGS SGR-1 (Green River Shale; 0.3 µg/g, certified value). As additional points of comparison we also measured [Hg] in the following reference materials: ATHO (Icelandic rhyolite obsidian), USGS AGV-2 (andesite from Guano Valley, Oregon), USGS BCR-2 (basalt from the Bridal Veil Flow Quarry near the Columbia River, Oregon), and USGS BHVO-2 (surface pahoehoe lava from the Halemaumau crater, Hawaii). Initial comparisons of a sample unknown (MC-PRB-EXOb), reference materials, and proposed carbonate concentration standards were digested in a range of acid normalities from 4N to concentrated solutions of both $HNO_3$ and Lefort aqua regia (also referred to as "inverse aqua regia," 3:1 $HNO_3$:HCl) to ensure that Hg dissolution was consistent and to test for any loss by volatilization through the acid reaction (none was observed). Splits of SRM-88b, NBS-20, and MC-PRB-EXOb were measured in aliquots ranging from 10 to 300 mg, and were digested in 3 ml of 4N trace metal grade $HNO_3$ in capped 7 ml acid-cleaned polytetrafluoroethylene (PTFE) or perfluoroalkoxy alkane polymer (PFA) vials to assess any potential matrix effects (e.g., the influence of impure sample matrix to enhance or suppress analytical detection) in these materials. To minimize sample material consumption, carbonate sample unknowns were routinely dissolved in 3 ml 4N trace metal grade $HNO_3$ in 7 ml PTFE/PFA vials with sample loads between 50 and 150 mg, and all non-carbonate material was routinely digested in 2 ml of Lefort aqua regia with identical sample loads. We tested the scalability of larger sample loads in acid digestions for future isotopic analyses with up to 500 mg of NBS-20 and 1000 mg of MESS-3 in 16 ml of Lefort aqua regia in 180 ml PTFE/PFA vials, later diluted by 50% with 16 ml 18.2 M$\Omega$ deionized water prior to analysis. Digestions were conducted at 80 °C between 12 and 48 h, after which an aliquot of the acid digestate (50–200 µl) was diluted with 5 ml of a 1% BrCl and 40 µl (NH₃OH)Cl solution, before being loaded onto a Nippon Instruments Inc. RA-3000FG + cold vapor atomic fluorescence spectroscopy (CV-AFS) analyzer for [Hg] determination in accordance with US EPA Method 1631[50] at the University of Michigan Biogeochemistry and Environmental Isotope Geochemistry Laboratory. Precision of the CV-AFS was determined from

an average measured analytical blank value of Hg at 0.070 ± 0.466 pg (6σ, *n* = 117) and an average [Hg] of 0.014 ± 0.093 pg g⁻¹ (6σ, *n* = 117), which would imply an analytical sensitivity for whole-sample [Hg] at effectively 0.1 pg g⁻¹. However, sample reproducibility both within a given analytical session and between analytical sessions on the AFS varied on average by 11% (as determined from the 2σ uncertainty of 22 replicated samples) of the calculated [Hg] from the measured mass of Hg. All samples replicates are listed in the extended data files. Process blanks averaged 2.50 ± 0.19 pg g⁻¹ (2σ, *n* = 12) as a definitive background, and for all determined [Hg] represents <0.5% of blank contribution to CV-AFS peak signal intensity, and in the majority of cases <0.025% blank contribution. For the purposes of clarity, all samples described as "replicates" can further be subdivided between samples from the same analytical session where the "date of acid digestion" of the solution permits comparison of aliquots from the same digestate solution, and "date of analysis" allows the comparison of aliquots run on different analytical sessions.

**Reference material mercury concentrations.** In addition to Carrara marble, we also analyzed [Hg] in other carbonate and non-carbonate reference materials in order to provide additional method development with respect to conducting sample digestions under varying acid strengths, temperatures for dissolution, and duration of acid reaction. We used the analysis of these reference materials as a 'proof of concept' approach to understanding Hg in high-temperature diagenesis and volcanism, which can inform the limits of preservation of Hg in carbonate fossils and the introduction of Hg to the environment volcanically.

NBS-20 (69.6 ± 7.1 ng g⁻¹) and COQ-1 (37.4 ng g⁻¹) yielded the highest measured [Hg]. NBS-20 (a sample of Solnhofen limestone) has been exhausted and is no longer commercially available, but COQ-1 can still be obtained from the USGS. COQ-1, a 120 Ma calcite-rich carbonatite from the Oka complex in Canada[51], bears surprisingly high [Hg] compared to other lava samples (see Supplementary Data File 1 and Supplementary Fig. 5), and we would anticipate Hg⁰ and Hg(II)-bearing mineral phase thermal decomposition to occur at magmatic temperatures. The only modern active carbonatite eruptions occur at Oldoinyo Lengai (Tanzania), where alkaline natrocarbonatite (Na- and K-rich) is produced at eruptive temperatures ranging from ~490 to 545 °C[52]. These eruptions are several hundred degrees cooler than all comparative modern silicate lavas and have lower average viscosities than modern basalts[51,52]. The [Hg] of COQ-1 compares to the [Hg] of reference materials that are also flow units of eruptive lavas including: BCR-2 (1.8 ng g⁻¹), BHVO-2 (3.0 ng g⁻¹), AGV-2 (2.8 ng g⁻¹), and ATHO (2.0 ng g⁻¹). BHVO-2 is a modern Hawaiian basalt, BCR-2 (basalt) and AGV-2 (andesite) are Cenozoic volcanics from Oregon, and ATHO is a Cenozoic rhyolite obsidian from Iceland. With an order of magnitude lower [Hg] values than COQ-1, these lavas may exhibit a greater effectiveness at thermally decomposing Hg(0) and Hg(II) phases (and/or exhibit higher volatile loss before these phases can crystallize). Fundamental compositional differences and/or postdepositional alteration could explain the discrepancies in these materials, or the crystallization of a higher proportion of Hg-retentive mineral phases in the lower temperature carbonatite melt prior to eruption (possibly due to less volatile loss of Hg). We anticipate that Hg-retentive mineral phases will be uncommon in volcanic settings due to the high degree of Hg⁰ volatilization and likely thermal decomposition of Hg (II)-bearing mineral phases from the generation of melt. The presence of significant [Hg] in all measured reference materials, carbonate and non-carbonate, shows promise in the determination of [Hg] throughout the geological record across a broad range of preserved materials. The retention of relatively elevated [Hg] in COQ-1 would also reinforce the idea that Hg is likely not easily mobilized diagenetically at elevated temperatures.

**Clumped isotope methodology.** We have utilized the carbonate clumped isotope paleothermometer from measured $\Delta_{47}$ values to determine the temperature of formation in which the fossil mollusk taxa from this study have precipitated their shells[27,28]. These formation temperatures are interpreted to represent coastal marine temperatures, where from the same sample aliquot we can obtain both [Hg] and temperatures directly reflecting the environment that these organisms resided in.

Aliquots of 3–5 mg per replicate of sampled biogenic carbonate powder were measured for δ¹⁸O, δ¹³C, and $\Delta_{47}$ isotopic compositions in the University of Michigan Stable Isotope Laboratory using the same instrumentation and procedure as previous studies[27,28], with a Porapak™ trap temperature held between −10 and −15 °C. Isotopic values were determined from measured voltage intensities and measured carbonate $\Delta_{47}$ values were placed in the absolute reference frame using heated (1000 °C) and H₂O-equilibrated (25 °C) standard gases, and converted to temperature values using acid fractionation factor of +0.067‰ and high-temperature composite calibration developed in the University of Michigan Stable Isotope Laboratory[53]. δ¹⁸O_{sw} values are calculated from carbonate δ¹⁸O and $\Delta_{47}$-derived temperatures using the appropriate fractionation factors for calcite and aragonite[27,28,53]. All taxa were dominantly calcite with the exception of *Anomia*, *Turritella, and Cyrtodaria* which were aragonitic. Measured δ¹⁸O, δ¹³C, $\Delta_{47}$, and calculated paleotemperature and δ¹⁸O_{sw} values for all samples are reported along with gas and carbonate standard data in the Supplementary Material.

Using the same approach as a previous study[28], we present raw data calculated with both old and new ¹⁷O parameters in Supplementary Data File 3 for future

recalculation of temperatures. Given that corrections to both measured unknowns and calibration samples within a given lab will likely be similar, we anticipate only small (<1–3 °C) variations between temperatures calculated using old and new $^{17}O$ parameters. Analytical uncertainties on single samples are reported in terms of 1 S.E. (determined on a minimum of three replicates per sample), and locality or region average temperatures are determined for multiple specimens, and reported with $1\sigma$ uncertainties.

**Strontium isotope analysis**. Carbonate samples were also analyzed for strontium isotopic compositions following methods established in previous studies[28]. Briefly, a split from each homogenous carbonate powder was digested in 7.5 M $HNO_3$ and Sr was separated using column chromatography with Eichrom Sr resin (after a previous study[28,54]). The Sr elutions collected from column separations were loaded onto rhenium filaments and measured for 200 cycles on either a Finnigan MAT 262 or Thermo Scientific Triton Plus$^{TM}$ TIMS at the University of Michigan. Measurement sessions where $^{87}Sr/^{86}Sr$ values of the standard NIST SRM-987 deviated from the accepted value of $^{87}Sr/^{86}Sr = 0.710248 \pm 0.000011$ were normalized to that value (see Supplementary Data File 1). The long-term mean $^{87}Sr/^{86}Sr$ value for NIST SRM-987 was $0.710238 \pm 0.000016$ ($1\sigma$).

An age for each measured fossil was calculated by comparison to the most recent iteration of the LOWESS global seawater strontium isotope curve for the Late Cretaceous obtained and applied in a previous study[28]. Sample $^{87}Sr/^{86}Sr$ values were matched to the closest $^{87}Sr/^{86}Sr$ value for the Campanian/ Maastrichtian portion of the mean LOWESS curve, and the analytical uncertainty of each strontium measurement was propagated through the uncertainty envelope of the LOWESS curve itself to provide the most conservative cumulative uncertainty on any given age (~0.45–1.5 Ma per sample).

## Data availability

The authors declare that the data supporting the findings of this study are available within the paper, three Supplementary Data Files (source data specifically for Figs. 2 and 4 can be found in Supplementary Data File 2), and in the Supplementary Information file.

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

## Acknowledgements
We thank L. Wingate for analytical assistance, D. Miller (University of Michigan Museum of Paleontology), K. McKinney (USGS), R. Nagendra (Anna University Chennai), J. Vellekoop (KU Leuven), and E. Clites (University of California Museum of Paleontology) for paleontological assistance and for access to sample collections. We thank R. Gabelman and B. Dzombak for their invaluable contributions in the field. We thank N. Hughes and E. Crowther for assistance with laboratory processing and modern sample collection. This work was funded by NSF-OCE-PRF #1420902, NSF-EAR #1123733, and the University of Michigan Scott Turner Award.

## Author contributions
K.W.M. performed the experiments with assistance from S.J.W., A.Y.K., J.D.G., M.W.J., S.V.P., and I.Z.W; K.W.M. and I.Z.W. conducted the field work; All authors contributed to the interpretations and analysis of the data; K.C.L., S.V.P., and J.D.B. supervised the project; all authors wrote the paper.

## Competing interests
The authors declare no competing interests.
