## [Peer Review File · Nature Communications]

Reviewers' comments:

Reviewer #1 (Remarks to the Author):

This manuscript presents novel records that demonstrate simultaneous increases in water temperature and Hg loading in coastal marine environments prior to the K-Pg boundary. The authors attribute these increases to the release of CO₂ and Hg⁰ by Deccan volcanism. Although there are many studies of sedimentary Hg (and Hg/TOC) across extinction boundaries, this is the first (of which I'm aware) to use carbonate from marine mollusks to reconstruct environmental Hg loads during a period of LIP volcanism and mass extinction. In addition, the coupling of the Hg record with a climate record sets this study apart from others in the existing literature.

Importantly, the authors are able to convincingly defend the validity of using marine mollusk carbonate to reconstruct environmental Hg by investigating Hg in modern and fossil carbonates from a number of globally-distributed localities. Their suggestion that global Hg loads during the late Maastrichtian may have been similar in magnitude to modern sites with anthropogenic legacy contamination is provocative and interesting.

One aspect of this study that I find particularly compelling is the addition of sedimentary [Hg] and Hg/TOC data to the record from Moscow Landing. The widespread (and often uncritical) use of Hg/TOC ratios to identify enhanced Hg loading to sediments in the geologic record has always made me a bit nervous -- I have often wondered if some of the "anomalies" are actually the result of low TOC contents rather than enhanced Hg loading. By demonstrating that sedimentary Hg/TOC records closely align with [Hg] in biogenic carbonates, this study provides a promising result in favor of using sedimentary Hg/TOC as an indicator of enhanced Hg deposition. However, regardless of this result, this study suggests to me that, where available, biogenic carbonates probably provide more reliable paleo-records of environmental Hg loads because they aren't complicated by variable TOC (or sulfur) contents.

This is an excellent study/manuscript and except for some very minor technical notes, I see no need for revision before publication.

A few very minor technical notes:

-The authors reference Fig. 4 and Fig. 5 in the manuscript (lines 163 and 174, respectively), but there are only two figures in the paper.

-Maastrichtian is misspelled on line 186

Reviewer #2 (Remarks to the Author):

This is a very fine, elegant scientific contribution for the understanding of the mass extinction at the K/Pg boundary using a new approach. Authors present the first record of measured Hg in mollusk shells paired with carbonate "clumped isotopes" composition ($\Delta 47$) aiming at investigating past marine T.

The results they reported from two localities (Seymour Island and Moscow Landing) are very stimulating as they point to abruptly increased in T and elevated Hg in the same specimen which not only reinforce the possible participation of volcanism but demonstrate that this approach used by the first time is very effective. This method needs confirmation applying it in well-known complete K/Pg sections (e.g. El Kef, Stevns Klint etc). It would have been suitable if authors had considered one of these sections in this study, instead of using (or besides using) the Moscow Landing section which is not complete (?) according to their description.

Recent dating of the Deccan event greatly improved our knowledge of the chronology of the several traps (Shoene et al., 2019 and Sprain et al., 2019, both published last February in

Science) and perhaps deserve to be mentioned here.

I strongly recommend this manuscript to be published in Nature Communications.

Alcides N. Sial

Reviewer #3 (Remarks to the Author):

Summary

The present manuscript presents results of clumped isotopes as a temperature proxy and [Hg] as a volcanism proxy from organic material which has been incorporated into bivalve shells that cross the K-Pg boundary. [Hg] concentrations from the latest Maastrichtian are compared with values from the Campanian and early Maastrichtian, as well as the Pleistocene and in modern environments, the latter which exhibit high concentrations of [Hg] due to anthropogenic inputs. Some latest Cretaceous shells contain [Hg] values that are higher than those from other pre-industrial shells and are more consistent with those from altered modern environments. These higher values are also associated with higher temperatures indicated by clumped isotopes. The data presented here suggest a close tie between Deccan volcanism and higher temperatures preceding the Chicxulub impact.

Data Quality

With regard to the main results, it is commendable that the authors extended their analyses to the early Maastrichtian in Antarctic samples, which provides excellent support for lower baseline values. However, it is surprising that sampling was not performed at a higher resolution in the main sampling interval (29R), especially since so much volatility is apparent in the five samples characterizing the main Deccan phase and the Chicxulub impact, (appears to be 1-2 samples per meter in Alabama and unknown distances for Antarctica). When discerning the influence of two major environmental events, more samples should be concentrated in that interval for several reasons – for example, to establish the duration/persistence of the high values.

Some problems are apparent in the establishment of the preindustrial/anthropogenic baselines, including diagenetic effects, the potential for taxon-specific [Hg] incorporation, and chronostratigraphy. One concerning aspect of the results is the “biostratigraphic” Maastrichtian values that are elevated, very much within the range of the Antarctic samples from the later Maastrichtian and they might be middle or late Maastrichtian – there are only a small number of samples that fall below the anthropogenic baseline value (and are used to define that range of preindustrial background and anthropogenic) – so to see such a range in values that could be from just about any point in the Maastrichtian is problematic. It would be difficult, on these data alone, to fully subscribe to the [Hg] boundaries described here. I would be more convinced if there was additional information characterizing the effect of diagenesis on [Hg] (since the authors refer to diagenetically altered specimens, which also have elevated [Hg] amounts). (As a side note, authors should probably include Sr concentration–fitting the values to the Sr LOWESS curve is okay, but that is a model – [Sr] and Sr/Mn values would provide additional, direct support for the influence of diagenesis, and I would assume that the authors have this information readily available). The importance of characterizing diagenesis is especially important for the Pleistocene bivalves, that presumably are shown to illustrate post-extinction pre-industrial values. The authors indicate that very little is known about the influence of diagenesis on [Hg], but that does not mean that its influence cannot be better characterized. Perhaps the minimal information about diagenesis could be overlooked, but there is considerable range in values, and there are only four points for the post-extinction preindustrial values that are critical for establishing that background. There are only six data points to establish the pre-extinction values – and I cannot find sedimentology/stratigraphy/taxon information about the Pee Dee in the supplemental materials (I can only assume that the sample from there is an *Exogyra* based on sample identifiers and context with other samples).

In going through the locality information, there is very little information about stratigraphy that

might be useful in contextualizing the [Hg] values the specimens produce, and the specimens sampled are the same species in each sample (there is sometimes an “unknown” species in the mix, but I must ignore those because I cannot discount that they are also from the same species perhaps simply abraded) – it would be instructive to see how [Hg] in different species from the same assemblage vary, because with the fairly low number of specimens used to define the preindustrial and anthropogenic values and the fact that the Cretaceous species are different from those, I cannot rule out some type of species-specific or genus-specific [Hg] range (uncertainty compounded because the process by which [Hg] enters the tissues is not known either.

Significance

The authors are overstating the relationship of their data to the end-Cretaceous mass extinction; Figure 1a and 1b indicate two extinction events, but no faunal data is included. If you look at the Schoene paper that the authors cite, the extinction events do not match – Schoene et al show an extinction event immediately prior to the boundary and a second extinction event in the Danian, while this manuscript shows an extinction at the K-Pg boundary and another extinction event in the late Maastrichtian. These could just be slightly offset, but the boundary is in different spots between these two papers, and since this is the source of the faunal data, this needs to be corrected. I attempted to see where the Schoene et al extinction rates came from, and was not able to determine that information from the manuscript or their supplemental data. Unfortunately, I cannot evaluate this component of the present manuscript because the Schoene et al data is from India and the present manuscript only indicates the extinction events on the Antarctic data. There is faunal turnover data from Antarctica that I am aware of – is the extinction information from that or some global dataset? Overall, the tie to extinction rates and, perhaps most importantly to other stratigraphic sections, is confusing, and to make the [Hg] and clumped isotope data relatable to the extinction event, some information about the extinction data itself is required.

The timing of the volcanism relative to the Chicxulub impact is fairly well-constrained based on absolute dates, flow volume estimates, biostratigraphy, etc. This manuscript presents an additional (and useful) proxy, but it is not clear how this defines the influence of the volcanism, though it is certainly instructive. Additional citations tying the present work to estimates of temperatures in the late Maastrichtian using carbon or oxygen isotopes (and other methods as well) should be included.

With regard to tying climate changes to volcanism, there certainly has been considerable work published on the climatic effects of the Deccan Traps, albeit using a different proxy than that employed by the authors to tie faunal changes to environmental shifts. Perhaps if this manuscript included faunal information tied to this newly applied proxy, this statement that the work present is the first would be more appropriate. This is an exciting tool that, when presented with clumped isotopes, does

Some line-by-line notes:

88: remove “the famed” – unnecessary characterization

Figure 1: The extinction events indicated are based on uncited data. Is this data in Seymour Island specifically? If so, what taxa? Why are extinction events not indicated in Moscow Landing chart?

Figure 2: Why no Danian taxa plotted? It would be compelling to include a separate plot of modern bivalve [Hg] values plotted against their known [Hg] water concentrations.

Reviewers' comments:

Reviewer #1 (Remarks to the Author):

This manuscript presents novel records that demonstrate simultaneous increases in water temperature and Hg loading in coastal marine environments prior to the K-Pg boundary. The authors attribute these increases to the release of CO₂ and Hg⁰ by Deccan volcanism. Although there are many studies of sedimentary Hg (and Hg/TOC) across extinction boundaries, this is the first (of which I'm aware) to use carbonate from marine mollusks to reconstruct environmental Hg loads during a period of LIP volcanism and mass extinction. In addition, the coupling of the Hg record with a climate record sets this study apart from others in the existing literature.

Importantly, the authors are able to convincingly defend the validity of using marine mollusk carbonate to reconstruct environmental Hg by investigating Hg in modern and fossil carbonates from a number of globally-distributed localities. Their suggestion that global Hg loads during the late Maastrichtian may have been similar in magnitude to modern sites with anthropogenic legacy contamination is provocative and interesting.

One aspect of this study that I find particularly compelling is the addition of sedimentary [Hg] and Hg/TOC data to the record from Moscow Landing. The widespread (and often uncritical) use of Hg/TOC ratios to identify enhanced Hg loading to sediments in the geologic record has always made me a bit nervous -- I have often wondered if some of the "anomalies" are actually the result of low TOC contents rather than enhanced Hg loading. By demonstrating that sedimentary Hg/TOC records closely align with [Hg] in biogenic carbonates, this study provides a promising result in favor of using sedimentary Hg/TOC as an indicator of enhanced Hg deposition. However, regardless of this result, this study suggests to me that, where available, biogenic carbonates probably provide more reliable paleo-records of environmental Hg loads because they aren't complicated by variable TOC (or sulfur) contents.

This is an excellent study/manuscript and except for some very minor technical notes, I see no

need for revision before publication.

We thank the reviewer for their comments.

A few very minor technical notes:

-The authors reference Fig. 4 and Fig. 5 in the manuscript (lines 163 and 174, respectively), but there are only two figures in the paper.

We have changed the figure references from “Fig. 4” to “Fig. 2,” and from “Fig. 5” to “Fig. S3, Supplementary Information,” accordingly.

-Maastrichtian is misspelled on line 186

Corrected.

Reviewer #2 (Remarks to the Author):

This is a very fine, elegant scientific contribution for the understanding of the mass extinction at the K/Pg boundary using a new approach. Authors present the first record of measured Hg in mollusk shells paired with carbonate “clumped isotopes” composition ($\Delta 47$) aiming at investigating past marine T.

The results they reported from two localities (Seymour Island and Moscow Landing) are very stimulating as they point to abruptly increased in T and elevated Hg in the same specimen which not only reinforce the possible participation of volcanism but demonstrate that this approach used by the first time is very effective. This method needs confirmation applying it in well-known complete K/Pg sections (e.g. El Kef, Stevns Klint etc). It would have been suitable if authors had considered one of these sections in this study, instead of using (or besides using) the Moscow Landing section which is not complete (?) according to their description.

We concur with Reviewer #2 that it would be desirable to perform [Hg] measurements on biogenic carbonates from other well-constrained K/Pg boundary sections, but contest that the ones we’ve studied are not well constrained.

The K-Pg boundary section at Seymour Island has been studied for over 3 decades (e.g. Macellari 1988) and is temporally well-constrained by magnetostratigraphy over multiple millions of years (Tobin et al., 2012), the presence of an Ir-anomaly to pinpoint the boundary (Elliot et al 1994), and $^{87}\text{Sr}/^{86}\text{Sr}$ records corroborating with global seawater model ages (MacArthur, 2001). The environmental change has been documented with extensive macro- and micro-fossil extinction patterns (Witts et al 2016), palynological records (Bowman et al., 2013), as well as stable and clumped isotopic records (Tobin et al., 2012; Petersen et al., 2016). The magnetostratigraphy in particular revealed consistent and high sedimentation rates leading to an expanded section of exposure, which allows for direct comparison between this site and others globally.

Although Moscow Landing is missing some time unconformably at the exact K/Pg boundary, it still preserved the key portions showing the large temperature increase aligning with the onset of the Deccan Traps. This section has been studied stratigraphically (Monroe and Hunt, 1958; Sohl, 1960; Smith, 1997), biostratigraphically (Moshkovitz and Habib, 1993; Larina et al., 2016), and isotopically (Meyer et al 2018). Smit et al. (1996) also document the presence of impact-related spherules at the base of channel deposits described as tsunami outflow deposits and defining the upper unconformable contact. We view this site as corroboration of the more-well-studied Seymour Island site.

We hope to expand this technique to other sites in the future, but would argue that these sites, the well-constrained Seymour Island site in particular, are sufficient to serve as ‘proof of concept’ and to form initial conclusions about the magnitude of environmental [Hg] loading from the Deccan traps as compared to modern anthropogenic background.

Bowman, V. C., Francis, J. E. & Riding, J. B. Late Cretaceous winter sea ice in Antarctica? Geology 41, 1227–1230 (2013).

Elliot, D. H., Askin, R. A., Kyte, F. T. & Zinsmeister, W. J. Iridium and dinocysts at the Cretaceous-Tertiary boundary on Seymour Island, Antarctica: implications for the KT event. Geology 22, 675–678 (1994).

Larina, E. et al. Upper Maastrichtian ammonite biostratigraphy of the Gulf Coastal Plain (Mississippi Embayment, southern USA). Cretaceous Research 60, 128–151 (2016).

Macellari, C. E. Stratigraphy, sedimentology, and paleoecology of Upper Cretaceous/Paleocene shelf-deltaic sediments of Seymour Island. 169, 25–53 (Geological Society of America Memoirs, 1988).

McArthur, J. M., Howarth, R. J. & Bailey, T. R. Strontium isotope stratigraphy: LOWESS version 3: Best fit to the marine Sr-isotope curve for 0-509 Ma and accompanying look-up table for deriving numerical age. Journal of Geology 109, 155–170 (2001).

Meyer, K. W., Petersen, S. V., Lohmann, K. C. & Winkelstern, I. Z. Climate of the Late Cretaceous North American Gulf and Atlantic Coasts. Cretaceous Research 89, 160–173 (2018).

Monroe, W. H. & Hunt, J. L. Geology of the Epes Quadrangle, Alabama. (1958).

Moshkovitz, S. & Habib, D. Calcareous nannofossil and dinoflagellate stratigraphy of the Cretaceous-Tertiary boundary, Alabama and Georgia. Micropaleontology 39, 167 (1993).

Sohl, N. F. Archeogastropoda, Mesogastropoda, and stratigraphy of the Ripley, Owl Creek, and Prairie Bluff Formations. (USGS Professional Paper 331-A, 1960).

Smit, J. et al. Coarse-grained, clastic sandstone complex at the K/T boundary around the Gulf of Mexico: Deposition by tsunami waves induced by the Chicxulub impact? Geological Society of America Special Papers 307, 151–182 (1996).

Smith, C. C. The Cretaceous-Tertiary Boundary at Moscow Landing, West-Central Alabama. Gulf Coast Association of Geological Societies Transactions 47, 533–540 (1997).

Tobin, T. S. et al. Extinction patterns, $\delta^{18}O$ trends, and magnetostratigraphy from a southern high-latitude Cretaceous–Paleogene section: Links with Deccan volcanism. Palaeogeography, Palaeoclimatology, Palaeoecology 350-352, 180–188 (2012).

Witts, J. D. et al. Macrofossil evidence for a rapid and severe Cretaceous-Paleogene mass extinction in Antarctica. Nat Comms 7, 1–9 (2016).

Recent dating of the Deccan event greatly improved our knowledge of the chronology of the several traps (Shoene et al., 2019 and Sprain et al., 2019, both published last February in Science) an perhaps deserve to be mentioned here.

We now include appropriate acknowledgement of these two studies as references #15 and #16.

I strongly recommend this manuscript to be published in Nature Communications.

We extend our thanks to the reviewer.

Reviewer #3 (Remarks to the Author):

Summary

The present manuscript presents results of clumped isotopes as a temperature proxy and [Hg] as a volcanism proxy from organic material which has been incorporated into bivalve shells that cross the K-Pg boundary. [Hg] concentrations from the latest Maastrichtian are compared with values from the Campanian and early Maastrichtian, as well as the Pleistocene and in modern environments, the latter which exhibit high concentrations of [Hg] due to anthropogenic inputs. Some latest Cretaceous shells contain [Hg] values that are higher than those from other pre-industrial shells and are more consistent with those from altered modern environments. These higher values are also associated with higher temperatures indicated by clumped isotopes. The data presented here suggest a close tie between Deccan volcanism and higher temperatures preceding the Chicxulub impact.

Data Quality

With regard to the main results, it is commendable that the authors extended their analyses to the early Maastrichtian in Antarctic samples, which provides excellent support for lower baseline values. However, it is surprising that sampling was not performed at a higher resolution in the main sampling interval (29R), especially since so much volatility is apparent in the five samples characterizing the main Deccan phase and the Chicxulub impact, (appears to be 1-2 samples per meter in Alabama and unknown distances for Antarctica). When discerning the influence of two major environmental events, more samples should be concentrated in that interval for several reasons – for example, to establish the duration/persistence of the high values.

We acknowledge Reviewer #3's point and agree that a higher sample density would be highly desirable in our time-series sites (Moscow Landing and Seymour Island), particularly for the chron 29R interval. However, our analyses are performed on macrofossils, which can only be found scattered through the relevant sections. Additionally, macrofossil shells, particularly at the Moscow Landing section, are exposed through seasonal erosion and at any given sampling interval only a limited number of specimens are accessible at any given time. This means resolution with this proxy method are by necessity of lower resolution than records of sediment [Hg], which can be performed at very high resolution. At both of our primary sample localities, macrofossil shells were recovered from every available horizon that could be discovered or exposed at the time of sampling. Multiple trips were conducted to find more material at the Moscow Landing exposure, with limited success.

Samples from other sites besides Seymour Island and Moscow Landing were obtained from museum collections (University of Michigan Museum of Paleontology, Ann Arbor, MI and University of California Museum of Paleontology, Berkeley, CA) and had little to no internal stratigraphic constraints, so were treated on a site-by-site basis.

Some problems are apparent in the establishment of the preindustrial/anthropogenic baselines, including diagenetic effects, the potential for taxon-specific [Hg] incorporation, and chronostratigraphy. One concerning aspect of the results is the “biostratigraphic” Maastrichtian values that are elevated, very much within the range of the Antarctic samples from the later Maastrichtian and they might be middle or late Maastrichtian – there are only a small number of samples that fall below the anthropogenic baseline value (and are used to define that range of preindustrial background and anthropogenic) – so to see such a range in values that could be from just about any point in the Maastrichtian is problematic. It would be difficult, on these data alone, to fully subscribe to the [Hg] boundaries described here.

First, we contend that the ‘pre-anthropogenic background’ is defined on values from 3 sites (Pee Dee River, South Carolina, Balsvik Quarry, Scania, Sweden; Rocky Bay, Bermuda), and does not rely on the “biostratigraphic” Maastrichtian samples.

Second, with the “biostratigraphic” Maastrichtian samples yielding from different parts within the Maastrichtian, the range of values observed would be consistent with some samples occurring within the Deccan Traps eruptive window and others not.

Lastly, we were targeting Late Maastrichtian in particular when searching for samples in the hopes of identifying and corroborating our more elevated from other sites, so despite poor age constraints, if anything these “biostratigraphic” sites are systematically biased towards late Maastrichtian.

I would be more convinced if there was additional information characterizing the effect of diagenesis on [Hg] (since the authors refer to diagenetically altered specimens, which also have elevated [Hg] amounts).

For the samples of biogenic carbonate measured for [Hg] in this study, there is only minor concern of [Hg] values being altered. We restricted ourselves to samples with recording surface Δ_{47} temperatures and $^{87}\text{Sr}/^{86}\text{Sr}$ compositions that best agreed with the LOWESS strontium

seawater curve (from McArthur, 2001) for the time period of consideration, with the former limiting burial temperatures to ≤ 100 °C. Analysis of high-temperature carbonates (Carrara marble and COQ-1, a carbonatite lava sample) suggests that diagenetic influence on marine carbonates would more likely result in the loss of Hg from a specimen with the exception of instances where native mercury or cinnabar deposits may be in close proximity to fossil deposits. We acknowledge the need for Hg-specific screening methods for diagenetic alteration, but since none have been developed yet, we used other proxy methods to screen our samples. We now present a significantly expanded discussion evaluating the potential for diagenetic [Hg] overprinting in section S12 and S13 of the Supplementary Information. Also included is the only, to our knowledge, documented instance of diagenetic cinnabar (HgS) biomineral replacement in rodent teeth.

The revisions to Supplementary Information, Section S12 are as follows:

“In making a first attempt at evaluating the role of diagenetic influences on [Hg] in biogenic carbonates, we determined [Hg] in Carrara marble. Carrara marble is a common geologic reference material that has been extensively used as a traditional $\delta^{18}\text{O}$ and $\delta^{13}\text{C}$ isotope standard. The boiling point of metallic Hg is 357 °C (Lide, 2004), and is readily capable of volatilizing at surface temperatures (e.g. evaporation from soils; Schlüter, 2000). Alternatively, Hg(II)-bearing phases such as mercury sulfides (HgS) and mercury oxides (HgO) have been demonstrated to have peak thermal decomposition temperatures (at surface pressures) of between 360 to 470 °C and 470 to 500 °C, respectively (Leckey and Nulf, 1994; Baláz and Godočikoyá, 2001). It has been suggested that Carrara marble has experienced peak metamorphic temperatures of 430 to 450 °C (Leiss and Molli, 2003) and fluid alteration at approximately 400 °C (Costagliola et al., 1999). Given the constraints on the metamorphic history of Carrara marble, we would not expect any metallic Hg^0 to be present in these samples, and only small quantities of Hg(II) mineral phases. We determined [Hg] in Carrara marble to be 1.4 ng g^{-1} , significantly elevated above average analytical blank values of $0.0079 \pm 0.021 \text{ ng g}^{-1}$, but only just outside the 'Pre-Industrial background' level. The retention of Hg in Carrara marble is encouraging for evaluating Hg in carbonates of various origins throughout the geological record given the possibility for Hg retention despite experiencing elevated metamorphic temperatures/conditions. For the samples of biogenic carbonate measured for [Hg] in this study, there is only minor concern of [Hg] values being altered (based on Δ_{47} and $^{87}\text{Sr}/^{86}\text{Sr}$ compositions with the former limiting burial temperatures to ≤ 100 °C), but we acknowledge the need for Hg-specific screening methods for diagenetic alteration. At present, we know of only one study that has quantified (via inductively-coupled plasma mass spectrometry) and documented fossilized biomineral replacement with cinnabar (HgS) filling-in the porous structure of rodent dentition, however this is acknowledged by the authors as exceedingly rare and the consequence of a nearby cinnabar deposits to the fossil locality (García-Alix et al., 2012). Additionally, only two recorded accounts of abiotic mercury carbonates have ever been documented, the polymorphs peterbaylissite and clearcreekite, $\text{Hg}_3^{1+}(\text{CO}_3)(\text{OH})\cdot 2\text{H}_2\text{O}$ (Roberts et al., 1995, 2001). Both minerals were discovered in the Clear Creek mercury mine of the New Idria district in San Benito County (California) and are described as being extremely rare and found in either immediate proximity to or co-occurring with cinnabar or native mercury (Roberts et al., 1995, 2001). Therefore, we believe that, while possible, Hg diagenetic influence on marine carbonates would more likely result in the loss of

Hg from a specimen with the exception of instances where native mercury or cinnabar deposits may be in close proximity to fossil deposits (none of our sites, to our knowledge)..”

The revisions to Supplementary Information, Section S13 are as follows:

“In addition to Carrara marble, we also analyzed [Hg] in other carbonate and non-carbonate reference materials in order to provide additional method development with respect to conducting sample digestions under varying acid strengths, temperatures for dissolution, and duration of acid reaction. We used the analysis of these reference materials as a ‘proof of concept’ approach to understanding Hg in high-temperature diagenesis and volcanism, which can inform the limits of preservation of Hg in carbonate fossils and the introduction of Hg to the environment volcanically.

NBS-20 ($69.6 \pm 7.1 \text{ ng g}^{-1}$) and COQ-1 (37.4 ng g^{-1}) yielded the highest measured [Hg]. NBS-20 (a sample of Solnhofen limestone) has been exhausted and is no longer commercially available, but COQ-1 can still be obtained from the USGS. COQ-1, a 120 Ma calcite-rich carbonatite from the Oka complex in Canada (Chen et al., 2013), bears surprisingly high [Hg] compared to other lava samples (see Supplementary Table S1), and we would anticipate Hg^0 and Hg(II)-bearing mineral phase thermal decomposition to occur at magmatic temperatures. The only modern active carbonatite eruptions occur at Oldoinyo Lengai (Tanzania), where alkaline natrocarbonatite (Na- and K-rich) is produced at eruptive temperatures ranging from ~ 490 to $545 \text{ }^\circ\text{C}$ (Krafft and Keller, 1989). These eruptions are several hundred degrees cooler than all comparative modern silicate lavas and have lower average viscosities than modern basalts (Krafft and Keller, 1989; Chen et al., 2013). The [Hg] of COQ-1 compares to the [Hg] of reference materials that are also flow units of eruptive lavas including: BCR-2 (1.8 ng g^{-1}), BHVO-2 (3.0 ng g^{-1}), AGV-2 (2.8 ng g^{-1}), and ATHO (2.0 ng g^{-1}). BHVO-2 is a modern Hawaiian basalt, BCR-2 (basalt) and AGV-2 (andesite) are Cenozoic volcanics from Oregon, and ATHO is a Cenozoic rhyolite obsidian from Iceland. With an order of magnitude lower [Hg] values than COQ-1, these lavas may exhibit a greater effectiveness at thermally decomposing Hg^0 and Hg(II) phases (and/or exhibit higher volatile loss before these phases can crystallize). Fundamental compositional differences and/or post-depositional alteration could explain the discrepancies in these materials, or the crystallization of a higher proportion of Hg-retentive mineral phases in the lower temperature carbonatite melt prior to eruption (possibly due to less volatile loss of Hg). We anticipate that Hg-retentive mineral phases will be uncommon in volcanic settings due to the high degree of Hg^0 volatilization and likely thermal decomposition of Hg(II)-bearing mineral phases from the generation of melt. The presence of significant [Hg] in all measured reference materials, carbonate and non-carbonate, shows promise in the determination of [Hg] throughout the geological record across a broad range of preserved materials. The retention of relatively elevated [Hg] in COQ-1 would also reinforce the idea that Hg is not likely not easily mobilized diagenetically at elevated temperatures.”

As a side note, authors should probably include Sr concentration—fitting the values to the Sr LOWESS curve is okay, but that is a model – [Sr] and Sr/Mn values would provide additional, direct support for the influence of diagenesis, and I would assume that the authors have this information readily available).

For the Seymour Island samples, [Sr], as well as other major and minor element concentrations (Fe, Mn, Mg) are published in Petersen et al. 2016. These were used to evaluate samples for

diagenetic alteration. Only the samples bearing the lowest major, minor, and trace elemental concentrations were utilized.

[Sr] is not available for other samples measured for $^{87}\text{Sr}/^{86}\text{Sr}$ compositions via TIMS as part of this study.

The importance of characterizing diagenesis is especially important for the Pleistocene bivalves, that presumably are shown to illustrate post-extinction pre-industrial values. The authors indicate that very little is known about the influence of diagenesis on [Hg], but that does not mean that its influence cannot be better characterized. Perhaps the minimal information about diagenesis could be overlooked, but there is considerable range in values, and there are only four points for the post-extinction preindustrial values that are critical for establishing that background.

These Pleistocene bivalves were collected from the same sites as gastropod fossils described in Winkelstern et al. (2017). This paper documents that the burial history and preservation of these shells is excellent. The range in [Hg] that is seen, therefore, can help constrain the expected variability within a single site at a single point in time. Although it looks similar to the variability in the Deccan sites, take note of the y-axis which is on the log scale and thus amplifies smaller differences between the low [Hg] samples. We suspect that this variability is not the result of diagenesis but actually more reflective of natural variability between and within individuals at a single point in time and may give a better estimate of uncertainty than our laboratory analysis blanks.

*Winkelstern, I. Z. et al. Meltwater pulse recorded in Last Interglacial mollusk shells from Bermuda. *Paleoceanography* 32, 132–145 (2017).*

There are only six data points to establish the pre-extinction values – and I cannot find sedimentology/stratigraphy/taxon information about the Pee Dee in the supplemental materials (I can only assume that the sample from there is an *Exogyra* based on sample identifiers and context with other samples).

The samples in question were described in more detail in another manuscript, Meyer et al. (2018; reference above), but the section and reference information has since been added to the Supplementary Information as follows:

Locality Identifier: BF-PD (“Burches Ferry”)

Formation: Peedee

Samples collected here:

- BF-PD-BELc: *Belemnitella americana* (d’Orbigny)
- BF-PD-EXOa: *Exogyra costata* (Say)
- BF-PD-EXOe: *Exogyra costata* (Say)

Location: Type section of the Pee Dee Belemnite carbon isotope standard reference material (PDB) at Burches Ferry, South Carolina (Ruffin, 1843; Siple, 1957; Swift, 1966).

Collection site description: Samples were collected from ~6.5 m of exposure of the Peedee formation from three distinct calcareous mudstone horizons bearing a mixed shell hash. Upper contact of the unit is unconformable. Refer to stratigraphic column in Meyer et al. (2018).

Collected by: K.W. Meyer, May 7th, 2015

Sample obtained from: collected by authors

The samples Reviewer #3 mentioned can now be easily cross-referenced and constitute two *Exogyra* samples and one *Belemnitella* sample.

In going through the locality information, there is very little information about stratigraphy that might be useful in contextualizing the [Hg] values the specimens produce...

The lack of stratigraphic information for many samples is because these were from museum collections, where availability of this type of information is dependent on who collected the samples initially and how they were archived. Many of these samples were originally collected decades ago, in some cases, dating back longer than a century (e.g. Dakhla samples – Hobbs, Jan. 1913). In many cases, the collectors were more concerned with documenting the specimen's existence without considering measuring stratigraphic position. This lack of good stratigraphic control at many sites is why we restricted our most in-depth comparisons to the two sites where we had the most precise stratigraphic information and samples that were unquestioningly collected *in situ*.

..., and the specimens sampled are the same species in each sample (there is sometimes an “unknown” species in the mix, but I must ignore those because I cannot discount that they are also from the same species perhaps simply abraded) – it would be instructive to see how [Hg] in different species from the same assemblage vary, because with the fairly low number of specimens used to define the preindustrial and anthropogenic values and the fact that the Cretaceous species are different from those, I cannot rule out some type of species-specific or genus-specific [Hg] range (uncertainty compounded because the process by which [Hg] enters the tissues is not known either).

We agree that a comprehensive interspecies comparison would be hugely beneficial to assess species- or genus-specific Hg partitioning. However, a few available inter and intra-species comparisons are possible within this dataset that can give some insight into this question.

In this study we provide samples from 5 different genera represented across modern, uncontaminated sites (Providence, Rhode Island; Lake Tahoe, California; Spectacle Island, Massachusetts), which all bear low [Hg] shell concentrations, and stand in contrast to the modern legacy-contaminated South River locality.

Additionally, two fossil specimens of different species from the same horizon (925m) in Seymour Island show consistent [Hg] of 5 and 6 ng/g, respectively. Belemnites specimens are represented from two pre-Deccan sites (Balsvik Quarry, Scania, Sweden; Pee Dee River, South Carolina) both show low [Hg]. These suggest that there are not dramatic species-specific differences in how Hg is incorporated into shell carbonate, at least across these 8+ genera that include marine and freshwater bivalves, belemnites, and gastropods.

Lastly, our conclusions that [Hg] were higher during the Deccan eruptive window than the Early Maastrichtian and Late Campanian included limited comparisons involving a single genus, *Exogyra*. We measured [Hg] in samples of *Exogyra* from Moscow Landing, Alabama and Kharga Oasis, Egypt (both with elevated Hg) and the Pee Dee River, South Carolina (low Hg). Therefore, if species-specific Hg incorporation/partitioning exists, we do not observe it.

Significance

The authors are overstating the relationship of their data to the end-Cretaceous mass extinction; Figure 1a and 1b indicate two extinction events, but no faunal data is included.

The two extinction events shown in Fig 1 have been discussed in-depth in Petersen et al. (2016), Tobin et al. (2012), Witts et al. (2016), and Tobin et al. (2017). As this faunal work is not part of this study, we cite the relevant papers and only indicate the points of mass extinction, not the faunal data itself.

The figure caption for Fig. 1 is now updated to reflect this.

Tobin, T. S., Bitz, C. M. & Archer, D. Modeling climatic effects of carbon dioxide emissions from Deccan Traps volcanic eruptions around the Cretaceous–Paleogene boundary. Palaeogeography, Palaeoclimatology, Palaeoecology 478, 139–148 (2017).

If you look at the Schoene paper that the authors cite, the extinction events do not match – Schoene et al show an extinction event immediately prior to the boundary and a second extinction event in the Danian, while this manuscript shows an extinction at the K-Pg boundary and another extinction event in the late Maastrichtian. These could just be slightly offset, but the boundary is in different spots between these two papers, and since this is the source of the faunal data, this needs to be corrected. I attempted to see where the Schoene et al extinction rates came from, and was not able to determine that information from the manuscript or their supplemental data. Unfortunately, I cannot evaluate this component of the present manuscript because the Schoene et al data is from India and the present manuscript only indicates the extinction events on the Antarctic data. There is faunal turnover data from Antarctica that I am aware of – is the extinction information from that or some global dataset? Overall, the tie to extinction rates and, perhaps most importantly to other stratigraphic sections, is confusing, and to make the [Hg] and clumped isotope data relatable to the extinction event, some information about the extinction data itself is required.

This comment has now been addressed with the correction made in the previous comment. The extinction in question was a local event described by Tobin et al. (2012). The placement of citations made our point unclear.

The timing of the volcanism relative to the Chicxulub impact is fairly well-constrained based on absolute dates, flow volume estimates, biostratigraphy, etc. This manuscript presents an additional (and useful) proxy, but it is not clear how this defines the influence of the volcanism, though it is certainly instructive.

We do not claim to ‘define the influence of volcanism’ with these results, but rather correlate samples that record climatically-relevant temperature variations with a reasonably distinct geochemical marker and to apply a quantification to environmental Hg loading from Deccan volcanism at that time period.

We found that the exceptionally high [Hg] values observed in the highly contaminated South River (Virginia) appear to be of the same magnitude of those measured in the samples known to correspond with the Deccan Traps eruptions, which is a profound statement considering the potential for Hg toxicity to high trophic positions organisms at such elevated [Hg]. We agree that the timing of certain events can be well constrained by absolute dates where dateable igneous rocks exist. The benefit of this technique is that it can be applied to macrofossils or marine

carbonates from locations far from datable lava flows and still constrain such samples to within or prior to the Deccan Traps eruptive window.

Additional citations tying the present work to estimates of temperatures in the late Maastrichtian using carbon or oxygen isotopes (and other methods as well) should be included.

Although we include Δ_{47} -temperature data in Fig 1, that is not the main focus of our study. Both of those records have been previously published elsewhere (Petersen et al., 2016; Meyer et al., 2018). Therefore, a full discussion comparing these records with other paleotemperatures can be found in those publications. Here we show them to relate local change in temps to global volcanism.

Nonetheless, the text of Figure 1 has been updated to reflect comparison to conventional $\delta^{18}\text{O}$ temperature estimates. The following citations (with global and temporal compilations in the references therein) have been added as per the Reviewer's request:

Steuber, T., Rauch, M., Masse, J.-P., Graaf, J. & Malkoč, M. Low-latitude seasonality of Cretaceous temperatures in warm and cold episodes. *Nature* **437**, 1341–1344 (2005).

Dutton, A., Huber, B. T., Lohmann, K. C. & Zinsmeister, W. J. High-resolution stable isotope profiles of a dimitobelid belemnite: Implications for paleodepth habitat and Late Maastrichtian climate seasonality. *PALAIOS* **22**, 642–650 (2007).

With regard to tying climate changes to volcanism, there certainly has been considerable work published on the climatic effects of the Deccan Traps, albeit using a different proxy than that employed by the authors to tie faunal changes to environmental shifts. Perhaps if this manuscript included faunal information tied to this newly applied proxy, this statement that the work present is the first would be more appropriate.

The “first” element of our work is not that we are the first to link climate change to volcanism, but rather that we are the first to measure the effects of climate change (temperature increase) and volcanism (Hg concentration increase) in a single specimen, allowing direct comparison and eliminating issues of bioturbation or cross-dating. We have clarified this section to make our point clearer and to avoid the appearance of invoking a first claim to the wrong assertion.

We have since expanded our discussion with the following paragraph:

When compared to global compilations of faunal changes during the Late Campanian through Late Maastrichtian, we see that our data agrees well with observed initial temperature increases (as determined through conventional $\delta^{18}\text{O}$ -derived temperature proxies) from 69.5 to 68 Ma, which also corresponds to events of significant evolutionary diversification⁴⁶. These compilations also find clustered extinctions of foraminifera and decreases in species richness in the same interval as our observed abrupt Δ_{47} -derived temperature excursion immediately prior to the K-Pg and during the Deccan Traps eruption interval⁴⁶.

We believe this now best reflects the Reviewer's suggestion and further strengthens the points made within the manuscript. This has been coupled with a citation to a global compilation of faunal change through this same time period here:

Keller, G., Punekar, J. & Mateo, P. Upheavals during the Late Maastrichtian: Volcanism, climate and faunal events preceding the end-Cretaceous mass extinction. *Palaeogeography, Palaeoclimatology, Palaeoecology* **441**, 137–151 (2016).

This is an exciting tool that, when presented with clumped isotopes, does
It seems that either this thought was left incomplete or was somehow lost during upload.

Some line-by-line notes:

88: remove “the famed” – unnecessary characterization
“Famed” was removed, as requested.

Figure 1: The extinction events indicated are based on uncited data. Is this data in Seymour Island specifically? If so, what taxa? Why are extinction events not indicated in Moscow Landing chart?

This comment has now been addressed in a previous comment with the amended citation and figure caption.

Figure 2: Why no Danian taxa plotted?

The Danian taxa have now been added to this figure under the subheading “Danian (Paleogene)”

It would be compelling to include a separate plot of modern bivalve [Hg] values plotted against their known [Hg] water concentrations.

The plot has been added to the Supplementary Information and is referenced as Figure S5 in the manuscript.

REVIEWERS' COMMENTS:

Reviewer #2 (Remarks to the Author):

I think authors provided elaborated responses to points that I have raised in my review of this manuscript and I am happy with the responses. Therefore, I think this manuscript may be published as it stands.

Reviewer #3 (Remarks to the Author):

In this version of the manuscript, the authors did a commendable job of addressing several of the concerns I mentioned in my first review. For example:

- more information about diagenesis was included
- more sample information was provided

However, my greatest problems were not addressed and, seemingly, cannot be addressed. For example:

- the lack of stratigraphic control
- the low number of samples in the interval of interest

I do understand that this manuscript is perhaps meant to be a proof-of-concept for a useful proxy being applied to a totally appropriate time interval. My issues mainly arise from the interpretation of the results, and what I see as a high likelihood of over-interpretation by those in the field of research that is concerned with the events of the latest Cretaceous. In that area, high-resolution stratigraphy is the convention, and should be considered more seriously by the authors when planning collections. Latest Cretaceous deposits are some of the mostly highly represented in the field of paleontology, at least for the Mesozoic, and reaching out to other researchers beyond the museum collections is not an unreasonable expectation.

In summary, the manuscript has been improved and includes more information that I would considering to be critical about samples. I would like to see more statements tempering the interpretations with regard to the K-Pg interval without additional samples or stratigraphic information, as they specifically relate to the issues I outlined above.

REVIEWERS' COMMENTS:

Reviewer #2 (Remarks to the Author):

I think authors provided elaborated responses to points that I have raised in my review of this manuscript and I am happy with the responses. Therefore, I think this manuscript may be published as it stands.

We thank Reviewer #2 for their input and suggestions to the manuscript.

Reviewer #3 (Remarks to the Author):

In this version of the manuscript, the authors did a commendable job of addressing several of the concerns I mentioned in my first review. For example:

- more information about diagenesis was included
- more sample information was provided

However, my greatest problems were not addressed and, seemingly, cannot be addressed. For example:

- the lack of stratigraphic control
- the low number of samples in the interval of interest

I do understand that this manuscript is perhaps meant to be a proof-of-concept for a useful proxy being applied to a totally appropriate time interval. My issues mainly arise from the interpretation of the results, and what I see as a high likelihood of over-interpretation by those in the field of research that is concerned with the events of the latest Cretaceous. In that area, high-resolution stratigraphy is the convention, and should be considered more seriously by the authors when planning collections. Latest Cretaceous deposits are some of the mostly highly represented in the field of paleontology, at least for the Mesozoic, and reaching out to other researchers beyond the museum collections is not an unreasonable expectation.

In summary, the manuscript has been improved and includes more information that I would considering to be critical about samples. I would like to see more statements tempering the interpretations with regard to the K-Pg interval without additional samples or stratigraphic information, as they specifically relate to the issues I outlined above.

As per Reviewer #3's suggestions we have further stressed the limitations and caveats involved in this study and the interpretations drawn from it as more suggestive from the limited data at-hand and as a first foray and pilot study for applying these geochemical methods to the samples we obtained. This is particularly emphasized in the final paragraph of the manuscript.

We maintain that between two university museum collections (UCMP, Berkeley; UMMP, Ann Arbor) we benefitted from over a century's worth of marine mollusk samples retrieved from across the globe. Whereas any study would be enhanced by more sample material and a larger dataset we are limited to the material that was loaned to us by these institutions and our collaborators. For example, I spent a week working through the invertebrate collections at the UCMP with the assistance of their Collections Manager, of all the material available some the largest proportion was immediately disqualified for analysis on the basis of extensive carbonate recrystallization or a lack of material for sampling (exhibiting as a mold or steinkern) -- thousands of candidate specimens were reduced to a few eligible for analysis that would also adhere closely to the museum's destructive analysis protocols.

We welcome future studies in applying these techniques and to see what is revealed by further analyses and evaluations of marine fossil material across the K-Pg boundary.